# Pooling by Sliced-Wasserstein Embedding

**Navid Naderializadeh**[*]
Department of Electrical and Systems Engineering
University of Pennsylvania
Philadelphia, PA 19104
nnaderi@seas.upenn.edu

**Joseph F. Comer, Reed W. Andrews, Heiko Hoffmann**
HRL Laboratories, LLC.
Malibu, CA 90265
{jfcomer, rwandrews, hhoffmann}@hrl.com

**Soheil Kolouri**[*]
Computer Science Department
Vanderbilt University
Nashville, TN 37235
soheil.kolouri@vanderbilt.edu

## Abstract

Learning representations from sets has become increasingly important with many applications in point cloud processing, graph learning, image/video recognition, and object detection. We introduce a geometrically-interpretable and generic pooling mechanism for aggregating a set of features into a fixed-dimensional representation. In particular, we treat elements of a set as samples from a probability distribution and propose an end-to-end trainable Euclidean embedding for sliced-Wasserstein distance to learn from set-structured data effectively. We evaluate our proposed pooling method on a wide variety of set-structured data, including point-cloud, graph, and image classification tasks, and demonstrate that our proposed method provides superior performance over existing set representation learning approaches. Our code is available at https://github.com/navid-naderi/PSWE.

## 1 Introduction

Many modern machine learning (ML) tasks deal with learning from set-structured data. In some cases, the input object itself is a set, as in point cloud classification/regression, and in other cases, the complex input object is described as a set of features after being through a backbone, i.e., a feature extractor. For instance, in graph mining, a graph is represented as a set of node embeddings, and in computer vision, an image is represented as a set of local features extracted from its different regions (i.e., fields of view). There are unique challenges in dealing with such set-structured data, namely: i) the set cardinalities could differ from one instance to another, and ii) the elements of the set do not necessarily have an inherent ordering. These challenges call for ML models that can both handle varied input sizes and are invariant to permutations, i.e., the model output does not change under any permutation of the input set elements.

Prior work on learning from set-structured data can be broadly categorized as methods based on either *implicit* or *explicit* embedding of sets into a Hilbert space. Implicit embedding approaches

---

[*]Work done while at HRL Laboratories, LLC.

35th Conference on Neural Information Processing Systems (NeurIPS 2021).

(i.e., kernel methods) rely on defining a distance/similarity measure (i.e., a kernel) between two sets [1, 2, 3, 4, 5, 6, 7, 8, 9]. These methods involve one of the two strategies of 1) solving a correspondence problem between elements of the input sets and measuring the similarity between corresponding elements, or 2) comparing all pairs of elements between the two sets based on a similarity measure (e.g., approaches based on Maximum Mean Discrepancy). On the other hand, explicit embedding methods learn a permutation-invariant mapping into a Hilbert space and provide a fixed-dimensional representation for a given input set that classic ML approaches could further process [10, 11, 12]. More recently, algorithms based on a composition of permutation-equivariant neural network backbones and permutation-invariant pooling mechanisms have been proposed to define a parametric permutation-invariant mapping [11, 13, 14, 15, 12, 16]. Notably, Zaheer et al. [11] proved that such a composition provides a universal approximator for any set function. Lee et al. [14] further showed that utilizing permutation-equivariant backbones that do not process set elements independently but model the interactions between the set elements (e.g., using self-attention) is theoretically and numerically advantageous. Similar observations have been made in the field of graph learning using various graph neural network (GNN) architectures [17, 18, 19]. In parallel, several works have studied the importance of permutation-invariant pooling mechanisms to go beyond the commonly used mean, sum, max, or similar operators [12, 13, 15, 16].

A convenient interpretation in dealing with sets is considering their elements as samples from an unknown underlying probability distribution and comparing/embedding these probability distributions to perform set learning. Due to this interpretation, optimal transport has played a prominent role in learning from sets. For instance, Kusner et al. [20] and later Huang et al. [21] represented a document as a set of words. They leveraged the *1-Wasserstein distance* (i.e., the earth mover's distance) to compare these sets with one another and define a measure of document similarity. Various researchers have devised similarly flavored approaches in computer vision by comparing images via calculating the Wasserstein distance between their sets of local features. For instance, Zhou et al. [22] use this distance to learn prototypes for image classes and perform few-shot inference, while Lin et al. [23] leverage it for designing diverse adversarial examples. More recently, similar ideas were used for image enhancement [24]. Finally, comparing sets via Wasserstein distances has also been proven to be useful in other applications including graph learning [9, 12, 16], domain adaptation [25, 26], and transfer learning [27].

In this work, we propose a novel theoretically-grounded and simple to compute permutation-invariant pooling mechanism for embedding sets of various sizes into a fixed-size representation. Our proposed method, which we refer to as Pooling by Sliced-Wasserstein Embedding (PSWE), provides an exact Euclidean embedding for the (generalized) sliced-Wasserstein (SW) distance. We start by defining a similarity measure between sets of samples based on the SW distance. We then propose an explicit set embedding for which the Euclidean distance between embedded sets equals the SW distance between them. In our experiments, we follow the recent work on set learning [11, 14] and use a permutation-equivariant backbone followed by our permutation-invariant pooling method to perform end-to-end learning on different data modalities. We demonstrate the scalability and effectiveness of our approach on various learning tasks, including point cloud classification, graph classification, and image recognition. Aside from introducing a novel pooling mechanism, one of the key numerical insights of our work is that basic pooling mechanisms, such as mean-pooling, provide competitive performance when the permutation-equivariant backbone is complex. However, for plain backbones (e.g., a shared multi-layer perceptron (MLP) among the set elements), more sophisticated pooling mechanisms, including our proposed PSWE method as well other recently-proposed pooling mechanisms in the literature (e.g., Pooling by Multi-Head Attention (PMA) [14] and Featurewise Sort Pool (FSPool) [15]) significantly boost the performance compared to basic pooling mechanisms.

## 2   Related Work

Permutation-invariant functions are crucial components in learning from sets and are often used as pooling layers to aggregate features from a set and provide a constant-size representation regardless of the set cardinality. Max, sum, and mean pooling are simple, yet very widely used, examples of such functions. Recently, various works have shown the effectiveness of more sophisticated and often learnable pooling operators in improving the performance of learning from set-structured data [13, 14, 28, 15, 10, 12]. Murphy et al. [13] introduced a pooling mechanism based on the average of a permutation equivariant-function applied to all re-orderings of the set elements. Summing over all

re-orderings of an input set is, of course, computationally prohibitive. Hence, one can use a canonical ordering of set elements (e.g., via sorting [15]), or learn to predict the optimal permutation for an input set [29, 30, 28].

Another prevalent idea is to perform pooling based on comparing an input set with trainable and fixed-size *reference* sets. For instance, Skianis et al. [10] proposed a pooling that consists of the distances between an input set and trainable reference sets, where the distance was calculated by solving the correspondence problems between the input and each reference set. More interestingly, this idea is analogous to pooling by multi-head attention (PMA), an important building block in the Set Transformer and Perceiver architectures [14, 31], where the cross-attention between trainable reference sets and an input set is used as a permutation-invariant function. Attention-based pooling [14, 32] has been shown to perform really well in practice on a wide range of applications.

We introduce a novel pooling mechanism by treating sets as empirical probability measures and calculating an embedding for these probability measures in which the Euclidean distance between two embedded sets is equal to the sliced-Wasserstein distance between their empirical distributions. Our work is closely related to the work by Mialon et al. [12] and Kolouri et al. [16]. In short, [12] proposes an approximate Euclidean embedding for the Wasserstein distance, similar to [16], in a reproducing kernel Hilbert space (RKHS), while our proposed framework is based on devising an exact Euclidean embedding for the (generalized) sliced-Wasserstein distance. Interestingly, our proposed pooling by sliced-Wasserstein embedding (PSWE) can also be viewed as a theoretically-grounded generalization of the sorting-based FSPool mechanism proposed in [15], where we show that the introduction of trainable slicers, as well as trainable reference sets, further boost the end-to-end performance in a wide spectrum of classification tasks.

## 3 Background

Let $X_i = \{x_n^i \in \mathbb{R}^d\}_{n=1}^{N_i}$ denote an input set with $N_i$ elements living in $\mathbb{R}^d$. We assume that the set elements are samples from an unknown underlying probability measure, $\mu_i$, defined in $\mathcal{X} \subseteq \mathbb{R}^d$ with probability density $d\mu_i(x) = p_i(x)dx$, and what we have observed is the empirical distribution $\hat{p}_i(x) = \frac{1}{N_i} \sum_{n=1}^{N_i} \delta(x - x_n^i)$, where $\delta(\cdot)$ is the Dirac delta function.

### 3.1 2-Wasserstein Distance

Let $\mu_i$ and $\mu_j$ denote two Borel probability measures with finite $2^{\text{nd}}$ moment defined on $\mathcal{X}_i, \mathcal{X}_j \subseteq \mathbb{R}^d$, with corresponding probability density functions $p_i$ and $p_j$, respectively. The 2-Wasserstein distance between $\mu_i$ and $\mu_j$ is the solution to the optimal mass transportation problem with $\ell_2$ transport cost [33]:

$$\mathcal{W}_2(\mu_i, \mu_j) = \left( \inf_{\gamma \in \Gamma(\mu_i, \mu_j)} \int_{\mathcal{X}_i \times \mathcal{X}_j} \|x_i - x_j\|^2 d\gamma(x_i, x_j) \right)^{\frac{1}{2}}, \tag{1}$$

where $\Gamma(\mu_i, \mu_j)$ is the set of all transportation plans $\gamma \in \Gamma(\mu_i, \mu_j)$ such that $\gamma(A \times \mathcal{X}_j) = \mu_i(A)$ and $\gamma(\mathcal{X}_i \times B) = \mu_j(B)$ for any Borel subsets $A \subseteq \mathcal{X}_i$ and $B \subseteq \mathcal{X}_j$. Due to Brenier's theorem [34], for absolutely continuous probability measures $\mu_i$ and $\mu_j$ (with respect to the Lebesgue measure), the 2-Wasserstein distance can be equivalently obtained from the Monge formulation [33],

$$\mathcal{W}_2(\mu_i, \mu_j) = \left( \inf_{f \in MP(\mu_i, \mu_j)} \int_{\mathcal{X}} \|x - f(x)\|^2 d\mu_i(x) \right)^{\frac{1}{2}}, \tag{2}$$

where $MP(\mu_i, \mu_j) = \{f : \mathcal{X}_i \to \mathcal{X}_j \mid f_{\#}\mu_i = \mu_j\}$ and $f_{\#}\mu_i$ represents the pushforward of measure $\mu_i$, characterized as $f_{\#}\mu_i(B) = \mu_i(f^{-1}(B))$ for any Borel subset $B \subseteq \mathcal{X}_j$. The mapping $f$ is referred to as a transport map [35], and the optimal transport map is called the Monge map. For discrete probability measures, when the transport plan $\gamma$ is a deterministic optimal coupling, such a transport plan is referred to as a Monge coupling [33]. In case of a non-deterministic transport plan $\gamma$, one can obtain an approximation of the Monge coupling via barycenteric projection, e.g., see [16, 12]. In this paper, we mainly use the 2-Wasserstein distance and hereafter, for brevity, we refer to it as the Wasserstein distance.

For one-dimensional probability measures, the Wasserstein distance has a closed-form solution and can be calculated as

$$\mathcal{W}_2(\mu_i, \mu_j) = \left( \int_0^1 |F_{\mu_i}^{-1}(\tau) - F_{\mu_j}^{-1}(\tau)|^2 d\tau \right)^{\frac{1}{2}}, \tag{3}$$

where $F_{\mu_i}^{-1}$ is the quantile function of $\mu_i$. The simplicity of calculating Wasserstein distances between one-dimensional probability measures has led to the idea of sliced-Wasserstein [36, 37, 38, 39] and generalized sliced-Wasserstein [40] distances, which we will review next.

### 3.2 (Generalized) Sliced-Wasserstein Distances

Let $g_\theta : \mathbb{R}^d \to \mathbb{R}$ be a parametric function with parameters $\theta \in \Omega_\theta \subseteq \mathbb{R}^{d_\theta}$, satisfying the regularity conditions in both inputs and parameters as presented in [40]. For sliced-Wasserstein distance, $g_\theta(x) = \theta^T x$ where $\theta \in \mathbb{S}^{d-1}$ is a unit vector in $\mathbb{R}^d$, and $\mathbb{S}^{d-1}$ denotes the unit $d$-dimensional hypersphere. The generalized slice of probability measure $\mu_i$ with respect to $g_\theta$ is the one-dimensional probability measure $g_{\theta\#}\mu_i$, which has the following density for all $t \in \mathbb{R}$,

$$p_i^\theta(t) := \int_{\mathcal{X}} \delta(t - g_\theta(x))d\mu_i(x). \tag{4}$$

The generalized sliced-Wasserstein distance is then defined as

$$\mathcal{GSW}_2(\mu_i, \mu_j) = \left( \int_{\Omega_\theta} \mathcal{W}_2^2(g_{\theta\#}\mu_i, g_{\theta\#}\mu_j)d\theta \right)^{\frac{1}{2}}. \tag{5}$$

Note that for $g_\theta(x) = \theta^T x$ and $\Omega_\theta = \mathbb{S}^{d-1}$, the generalized sliced-Wasserstein distance is equivalent to the sliced-Wasserstein distance. Equation (5) is the expected value of the Wasserstein distances between slices of distributions $\mu_i$ and $\mu_j$.

Extensions of the (generalized) sliced-Wasserstein distance include max (generalized) sliced-Wasserstein distance [38, 40], in which the expected value in (5) is substituted with a maximum over $\Omega_\theta$, i.e.,

$$\max\text{-}\mathcal{GSW}_2(\mu_i, \mu_j) = \max_{\theta \in \Omega_\theta} \mathcal{W}_2(g_{\theta\#}\mu_i, g_{\theta\#}\mu_j), \tag{6}$$

subspace-robust Wasserstein distance [41], which generalizes the notion of slicing to a projection onto subspaces, and the distributional sliced-Wasserstein distance [42] that proposes to replace the expectation with respect to the uniform distribution on $\Omega_\theta$ with a non-uniform and learnable distribution.

From an algorithmic point of view, the expectation in (5) is approximated using Monte-Carlo integration, which results in an average of a set of Wasserstein distances between random slices of $d$-dimensional measures. In practice, however, GSW distances only output a good Monte-Carlo approximation using a large number of slices, while max-GSW distances achieve similar results with only a single slice, although at the cost of an optimization over $\theta$.

## 4 PSWE: Pooling by Sliced-Wasserstein Embedding

### 4.1 (Generalized) Sliced-Wasserstein Embedding

We propose a Euclidean embedding for probability measures, such that the weighted Euclidean distance between two embedded measures is equivalent to the GSW distance between them. Consider a set of probability measures $\{\mu_i\}_{i=1}^M$ with densities $\{p_i\}_{i=1}^M$. For simplicity of notation, let $\mu_i^\theta := g_{\theta\#}\mu_i$ denote the slice of measure $\mu_i$ with respect to $g_\theta$. Also, let $\mu_0$ denote a reference measure, with $\mu_0^\theta$ representing its corresponding slice. Then, it is straightforward to show that the optimal transport map (i.e., Monge map) between $\mu_i^\theta$ and $\mu_0^\theta$ can be written as

$$f_i^\theta = F_{\mu_i^\theta}^{-1} \circ F_{\mu_0^\theta}, \tag{7}$$

where as mentioned before, $F_{\mu_i^\theta}^{-1}$ and $F_{\mu_0^\theta}^{-1}$ respectively denote the quantile functions of $\mu_i^\theta$ and $\mu_0^\theta$. Now, letting $id$ denote the identity function, we can write the so-called cumulative distribution transform (CDT) [43] of $\mu_i^\theta$ as

$$\nu_i^\theta := f_i^\theta - id. \tag{8}$$

For a fixed $\theta$, $\nu_i^\theta$ satisfies the following conditions, the proof of which can be found in the Supplementary Material:

C1: The weighted 2-norm of $\nu_i^\theta$ equals the Wasserstein distance between $\mu_i^\theta$ and $\mu_0^\theta$, i.e.,

$$\|\nu_i^\theta\|_{\mu_0^\theta,2} = \left( \int_{\mathbb{R}} \|\nu_i^\theta(t)\|_2^2 d\mu_0^\theta(t) \right)^{\frac{1}{2}} = \mathcal{W}_2(\mu_i^\theta, \mu_0^\theta),$$

hence implying that $\|\nu_0^\theta\|_{\mu_0^\theta,2} = 0$.

C2: the weighted $\ell_2$ distance between $\nu_i^\theta$ and $\nu_j^\theta$ equals the Wasserstein distance between $\mu_i^\theta$ and $\mu_j^\theta$, i.e.,

$$\|\nu_i^\theta - \nu_j^\theta\|_{\mu_0^\theta,2} = \mathcal{W}_2(\mu_i^\theta, \mu_j^\theta).$$

Finally, the GSW distance between two measures, $\mu_i$ and $\mu_j$, can be obtained as

$$\mathcal{GSW}_2(\mu_i, \mu_j) = \left( \int_{\Omega_\theta} \|\nu_i^\theta - \nu_j^\theta\|_{\mu_0^\theta,2}^2 d\theta \right)^{\frac{1}{2}} = \left( \int_{\Omega_\theta} \int_{\mathbb{R}} \|\nu_i^\theta(t) - \nu_j^\theta(t)\|_2^2 d\mu_0^\theta(t) d\theta \right)^{\frac{1}{2}}. \tag{9}$$

Based on (9), for probability measure $\mu_i$, the mapping to the embedding space is obtained via $\phi(\mu_i) := \{\nu_i^\theta \mid \theta \in \Omega_\theta\}$.

## 4.2 Algorithmic Considerations

In this section, we introduce our novel pooling algorithm, termed pooling by sliced-Wasserstein embedding (PSWE). Let $X_i = \{x_n^i \sim p_i\}_{n=1}^{N_i}$ denote an input set with $N_i$ elements, and $X_0 = \{x_n^0 \sim p_0\}_{n=1}^N$ denote the set of $N$ samples from a trainable reference set. Let $\Theta_L = \{\theta_l \sim \mathcal{U}_{\Omega_\theta}\}_{l=1}^L$ denote a set of $L$ parameters sampled uniformly from $\Omega_\theta$. Then, the empirical distribution of the $l^{\text{th}}$ slice of $p_i$ can be written as $\hat{p}_i^{\theta_l} = \frac{1}{N_i} \sum_{n=1}^{N_i} \delta(t - g_{\theta_l}(x_n^i))$. To obtain $\nu_i^{\theta_l}$, we need to calculate the Monge coupling between $\hat{p}_i^{\theta_l}$ and $\hat{p}_0^{\theta_l}$. In what follows, we consider two scenarios:

1. When the input set and the reference set have the same cardinalities, i.e., $N_i = N$, the Monge coupling (i.e., the discrete counterpart of the Monge map shown in (7)) is obtained by sorting $X_i^{\theta_l} := \{g_{\theta_l}(x_n^i)\}_{n=1}^{N_i}$ and $X_0^{\theta_l}$. Let $\pi_i[\cdot]$ denote the permutation indices (i.e., `argsort`) obtained by sorting $X_i^{\theta_l}$. Then, letting $\pi_0^{-1}$ denote the ordering that permutes the sorted set back to the original ordering based on sorting of elements in $X_0^{\theta_l}$, the Monge coupling is obtained via $\pi_i[\pi_0^{-1}[\cdot]]$ and the per-slice embedding is calculated as

$$[\nu_i^{\theta_l}]_n = g_{\theta_l}\left( x_{\pi_i[\pi_0^{-1}[n]]}^i \right) - g_{\theta_l}(x_n^0). \tag{10}$$

2. When the set cardinalities vary, the Monge coupling can be obtained via interpolation using (7). In our experiments, we use the PyTorch implementation of linear interpolation[2] to evaluate $F_{\mu_i^{\theta_l}}^{-1}$. The per-slice embedding is calculated as

$$[\nu_i^{\theta_l}]_n = F_{\mu_i^{\theta_l}}^{-1}\left( \frac{\pi_0^{-1}[n] + 1}{N} \right) - g_{\theta_l}(x_n^0), \tag{11}$$

where $F_{\mu_0^{\theta_l}}(x_n^0) = \frac{\pi_0^{-1}[n]+1}{N}$, assuming that the indices start from 0.

Note that, regardless of the cardinality of the input set, the per-slice embedding is $N$-dimensional, i.e., $\nu_i^{\theta_l} \in \mathbb{R}^N$. The final embedding is then defined as $\phi(\mu_i) = [\nu_i^{\theta_1}, ..., \nu_i^{\theta_L}] \in \mathbb{R}^{N \times L}$, which satisfies

$$\mathcal{GSW}_2(\mu_i, \mu_j) \approx \|\phi(\mu_i) - \phi(\mu_j)\|_F, \tag{12}$$

where $\|\cdot\|_F$ denotes the Frobenius norm, and the approximation is due to the Monte-Carlo integral approximation with the $L$ slices.

---

[2] https://github.com/aliutkus/torchinterp1d

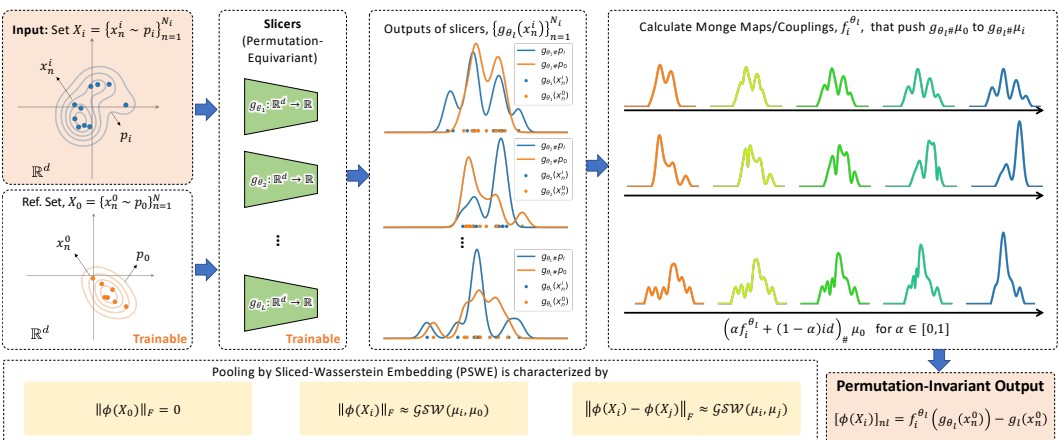

Figure 1: An overview of the proposed PSWE method. Each $d$-dimensional element in a given input set $X_i$, as well as each element in the trainable reference set $X_0$ is passed through multiple trainable slicers $\{g_{\theta_l}\}_{l=1}^{L}$. For each slicer, we then perform interpolation on the slicer outputs and derive the optimal transport maps that push the slicer output distributions of the reference set to the slicer output distributions of a given set via (7), (10), and (11). The resultant transport maps are then concatenated across all slices to derive the final set embedding.

### 4.3 On Projection Complexity of Sliced Distances

Given the high-dimensional nature of the problems of interest in machine learning, and the fact that samples often live on a low-dimensional manifold, one requires a large number of random projections, $L$, to obtain a good approximation of the GSW distance. This issue is related to the projection complexity of the sliced distances [38, 40]. Given the dependence of our pooling dimensionality on the number of slices, $L$, we would like to avoid using very large numbers of slices. Here, we devise a unique approach that ties our proposed embedding to metric learning. First, we note that ideas like max-GSW [40, 38] or subspace-robust Wasserstein distance [41] would not be practical in this setting, as the slicing parameters, $\Theta_L$, are fixed for all probability measures and not chosen separately for each pairs of probability measures $(\mu_i, \mu_j)$.

Given the training input sets, i.e., $\{X_i\}_{i=1}^{M}$, and a reference set, $X_0$, we seek an optimal set of $L$ slices $\Theta_L^*$ that could be learned from the data alongside the other parameters in an end-to-end manner. This idea is related to [42] as it is similar to learning a distribution over the unit hypersphere from which we are sampling our $L$ slices. The optimization on $\Theta_L^*$ ties the PSWE framework to the field of metric learning, allowing us to find slices or, in other words, an *embedding* with a specific statistical characterization.

To put it all together, our pooling requires identifying: 1) the type of slicer $g_\theta : \mathbb{R}^d \to \mathbb{R}$ (e.g., $g_\theta(x) = \theta^T x$), 2) the number of slices, $L$, and 3) the number of elements in the reference set, $N$. Then, for an input set $X_i$ with $N_i$ elements, PSWE first slices the elements of the input and reference sets with respect to slicers $g_{\theta_l}$ for $l \in \{1, ..., L\}$. Then, it sorts the sliced values $\{g_{\theta_l}(x_n^i)\}_{n=1}^{N_i}$ and $\{g_{\theta_l}(x_n^0)\}_{n=1}^{N}$ and calculates or approximates the corresponding Monge couplings according to (10) or (11), respectively. Finally, PSWE calculates the per-slice embedding $\nu_i^{\theta_l}$ and returns $\phi(X_i) = [\nu_i^1, ..., \nu_i^L] \in \mathbb{R}^{N \times L}$. This procedure is depicted in Figure 1, as well as Algorithm 1. Note that in our proposed framework, the slicer parameters and the reference set elements are all trainable parameters that are updated using backpropagation of gradients due to the objective function of interest.

## 5 Experimental Evaluation

We evaluate the proposed PSWE method on a variety of point cloud, graph, and image datasets as depicted in Figure 2. For comparison, we consider four different pooling methods: Global average pooling (GAP), global max pooling (MAX–evaluated on image classification only), Pooling by

---

**Algorithm 1** Pooling by Sliced Wasserstein Embedding

---

**procedure** $\text{PSWE}(X_i = \{x_n^i \in \mathbb{R}^d\}_{n=1}^{N_i})$

    **Trainable parameters:** Slicer parameters $\Theta_L \in \mathbb{R}^{d_\theta \times L}$, Reference elements $X_0 \in \mathbb{R}^{N \times d}$

    **for** $l = 1$ to $L$ **do**

        Calculate $g_{\theta_l}(X_i) := \{g_{\theta_l}(x_n^i)\}_{n=1}^{N_i}$ and $g_{\theta_l}(X_0) = \{g_{\theta_l}(x_n^0)\}_{n=1}^{N}$

        Calculate $\pi_i = \text{argsort}(g_{\theta_l}(X_i))$, $\pi_0 = \text{argsort}(g_{\theta_l}(X_0))$, and $\pi_0^{-1}$

        **if** $N_i = N$ **then**

            Calculate $\nu_i^{\theta_l}$ according to (10)

        **else**

            Calculate $\nu_i^{\theta_l}$ according to (11)

    **return** $\phi(X_i) = [\nu_i^{\theta_1}, ..., \nu_i^{\theta_L}] \in \mathbb{R}^{N \times L}$

---

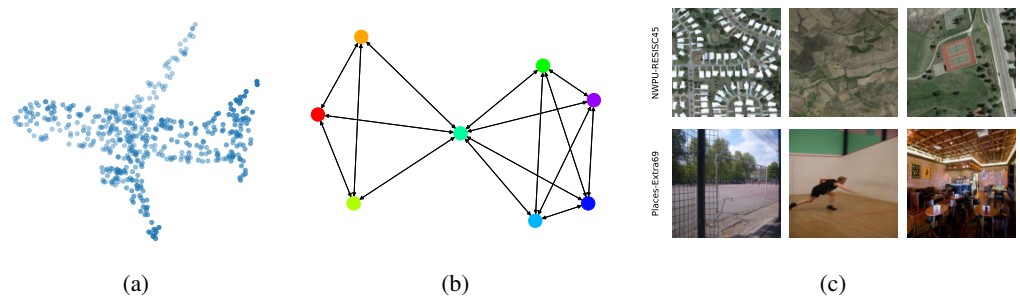

      (a)                  (b)                  (c)

Figure 2: We evaluate the performance of PSWE and other baseline pooling methods on (a) 3D point cloud classification from ModelNet40 dataset [44], (b) TUD graph classification datasets [45], and (c) image recognition on NWPU-RESISC45 [46] and Places-Extra69 [47] datasets.

Multi-head Attention (PMA) [14], and Featurewise Sort Pooling (FSPool) [15]. In all the PSWE experiments, to ease the optimization process of reference elements, we optimize the reference elements at the output of the slicers rather than in the input space of the slicers. Moreover, for both PMA and PSWE, to reduce the embedding size, we use a similar weighting approach to that of FSPool, where the output embedding (which is of size $N \times L$ and $N \times d$ for PSWE and PMA, respectively) is multiplied elementwise by a learnable weight matrix $W$ of the same size, and the result is summed over the rows to derive a final $L$-dim and $d$-dim embedding for PSWE and PMA, respectively. Further details on the experiments can be found in the Supplementary Material.

## 5.1 Point Cloud Processing

We consider the ModelNet40 dataset [44], consisting of 3-dimensional point clouds derived from triangular meshes of 12,311 CAD models belonging to 40 object categories. We sample 1024 points uniformly at random from each object as in [48, 49] and use the official split, with 9,843 training samples and 2,468 test samples. We consider two different backbones, namely multi-layer perceptron (MLP) and induced set attention block (ISAB) from the Set Transformer architecture [14].

Table 1 shows the test accuracy achieved by the proposed PSWE method using different numbers of slices ($L \in \{1, 4, 16, 64, 256, 1024\}$) and the baseline pooling methods of GAP, PMA, and FSPool. As the table shows, for both backbone types, PSWE is able to outperform other pooling methods for high-enough numbers of slices. Furthermore, it is noteworthy that while mean-pooling does not perform well when using an MLP backbone, performing message passing among the set elements using the ISAB backbone significantly boosts GAP's performance, suggesting that simple averaging of the per-element embeddings suffices to achieve a high performance level. This implies that there is an inherent trade-off between the backbone and pooling complexity, and to maintain a high accuracy level, at least one of the two components should be complex enough. A comparison of the wall-clock training and testing times of PSWE and the baseline pooling methods on the ModelNet40 dataset, as well as experimental results on visualizing the closest and farthest samples to/from the trained reference for PSWE, can be found in the Supplementary Material.

| Backbone | GAP | PMA | FSPool | PSWE | | | | | |
|---|---|---|---|---|---|---|---|---|---|
| | | | | $L=1$ | $L=4$ | $L=16$ | $L=64$ | $L=256$ | $L=1024$ |
| MLP | $57.8 \pm 0.5$ | $86.6 \pm 0.6$ | $85.8 \pm 0.5$ | $14.9 \pm 1.0$ | $52.9 \pm 2.1$ | $77.4 \pm 0.4$ | $83.9 \pm 0.6$ | $86.5 \pm 0.5$ | $86.9 \pm 0.3$ |
| ISAB | $86.6 \pm 0.5$ | $87.6 \pm 0.6$ | $87.3 \pm 0.5$ | $32.4 \pm 3.6$ | $83.9 \pm 0.6$ | $86.2 \pm 0.5$ | $86.9 \pm 0.3$ | $87.3 \pm 0.4$ | $87.6 \pm 0.4$ |

Table 1: Test classification accuracy (%) of the proposed PSWE method and the baseline pooling mechanisms on the ModelNet40 point cloud dataset using multi-layer perceptron (MLP) and induced set attention block (ISAB) [14] backbones.

| | Backbone | GAP | PMA | FSPool | PSWE | | | | | |
|---|---|---|---|---|---|---|---|---|---|---|
| | | | | | $L=1$ | $L=4$ | $L=16$ | $L=64$ | $L=256$ | $L=1024$ |
| IMDB-B | GCN | $69.6 \pm 3.9$ | $74.1 \pm 5.3$ | $75.5 \pm 3.7$ | $72.6 \pm 7.6$ | $74.6 \pm 5.8$ | $\mathbf{77.3 \pm 5.1}$ | $73.0 \pm 7.5$ | $72.7 \pm 5.1$ | $73.5 \pm 6.1$ |
| IMDB-B | GAT | $73.4 \pm 3.5$ | $70.5 \pm 7.6$ | $72.4 \pm 6.9$ | $71.3 \pm 7.2$ | $\mathbf{74.4 \pm 5.8}$ | $74.0 \pm 5.7$ | $70.9 \pm 7.6$ | $73.0 \pm 3.6$ | $73.4 \pm 6.0$ |
| IMDB-B | GIN | $73.0 \pm 5.8$ | $70.0 \pm 8.0$ | $73.4 \pm 6.1$ | $73.8 \pm 6.5$ | $72.5 \pm 6.0$ | $72.0 \pm 4.3$ | $72.3 \pm 8.1$ | $\mathbf{74.6 \pm 7.0}$ | $68.8 \pm 5.1$ |
| IMDB-M | GCN | $\mathbf{51.8 \pm 4.2}$ | $50.1 \pm 2.8$ | $51.1 \pm 5.4$ | $44.2 \pm 4.9$ | $50.7 \pm 4.6$ | $50.8 \pm 3.8$ | $49.6 \pm 3.6$ | $51.4 \pm 4.2$ | $50.2 \pm 4.7$ |
| IMDB-M | GAT | $49.7 \pm 3.4$ | $49.6 \pm 4.9$ | $\mathbf{50.2 \pm 3.9}$ | $44.3 \pm 4.2$ | $49.2 \pm 4.1$ | $50.2 \pm 4.5$ | $48.2 \pm 4.6$ | $47.9 \pm 4.3$ | $49.4 \pm 4.5$ |
| IMDB-M | GIN | $49.7 \pm 2.9$ | $50.2 \pm 3.0$ | $\mathbf{50.8 \pm 5.3}$ | $44.6 \pm 4.4$ | $49.1 \pm 2.7$ | $48.0 \pm 6.4$ | $50.6 \pm 2.6$ | $50.2 \pm 3.4$ | $49.5 \pm 4.0$ |
| RDT-B | GCN | $81.9 \pm 2.6$ | $82.2 \pm 2.5$ | $\mathbf{84.0 \pm 2.8}$ | $77.5 \pm 4.5$ | $80.4 \pm 3.3$ | $81.5 \pm 2.1$ | $82.1 \pm 3.2$ | $81.9 \pm 3.2$ | $81.7 \pm 3.0$ |
| RDT-B | GAT | $75.8 \pm 3.3$ | $76.0 \pm 3.5$ | $\mathbf{84.7 \pm 3.3}$ | $78.7 \pm 2.5$ | $82.0 \pm 3.2$ | $82.1 \pm 3.1$ | $81.7 \pm 3.2$ | $83.0 \pm 3.3$ | $81.7 \pm 3.7$ |
| RDT-B | GIN | $81.2 \pm 3.1$ | $77.6 \pm 7.9$ | $84.4 \pm 2.9$ | $83.2 \pm 3.3$ | $83.1 \pm 2.8$ | $83.8 \pm 2.8$ | $\mathbf{84.6 \pm 2.3}$ | $83.9 \pm 2.8$ | $83.7 \pm 1.7$ |
| PROTEINS | GCN | $69.1 \pm 5.2$ | $72.4 \pm 5.9$ | $\mathbf{74.9 \pm 5.4}$ | $72.5 \pm 3.9$ | $73.3 \pm 5.3$ | $73.3 \pm 5.6$ | $73.2 \pm 6.1$ | $72.8 \pm 6.0$ | $73.9 \pm 4.6$ |
| PROTEINS | GAT | $69.7 \pm 4.4$ | $72.4 \pm 6.1$ | $73.0 \pm 5.1$ | $72.9 \pm 4.6$ | $73.1 \pm 4.5$ | $72.8 \pm 4.4$ | $73.9 \pm 4.6$ | $\mathbf{74.4 \pm 4.4}$ | $73.7 \pm 5.5$ |
| PROTEINS | GIN | $69.8 \pm 6.5$ | $72.3 \pm 4.7$ | $72.6 \pm 4.5$ | $71.3 \pm 4.9$ | $72.4 \pm 6.0$ | $73.4 \pm 4.8$ | $73.5 \pm 4.4$ | $73.0 \pm 5.0$ | $\mathbf{74.9 \pm 3.9}$ |
| ENZYMES | GCN | $25.0 \pm 5.1$ | $32.1 \pm 4.5$ | $33.5 \pm 4.2$ | $20.0 \pm 3.9$ | $24.9 \pm 6.5$ | $31.8 \pm 5.1$ | $32.5 \pm 3.0$ | $\mathbf{37.8 \pm 4.9}$ | $33.7 \pm 3.9$ |
| ENZYMES | GAT | $24.2 \pm 5.3$ | $28.8 \pm 3.9$ | $34.2 \pm 6.7$ | $22.3 \pm 4.0$ | $26.3 \pm 5.6$ | $30.6 \pm 4.7$ | $34.6 \pm 3.2$ | $\mathbf{38.1 \pm 5.5}$ | $34.9 \pm 3.9$ |
| ENZYMES | GIN | $29.6 \pm 6.3$ | $30.1 \pm 4.8$ | $43.6 \pm 6.1$ | $19.1 \pm 5.5$ | $25.9 \pm 4.6$ | $36.5 \pm 3.1$ | $37.2 \pm 5.7$ | $\mathbf{45.4 \pm 7.0}$ | $40.0 \pm 6.0$ |

Table 2: Cross-validation accuracy (%) of PSWE with different numbers of slices, as well as baseline pooling methods on different TUD graph classification tasks [45] using three backbones of GCN [17], GAT [18], and GIN [19]. The best performing pooling method in each row (i.e., (dataset, backbone) pair) is highlighted in **bold**.

## 5.2 Graph Classification

Next, we consider the prominent TUD benchmark [45] and evaluate the performance of the proposed method on five graph classification datasets, consisting of social network (IMDB-B, IMDB-M, REDDIT-B) and bio-informatics (ENZYMES, PROTEINS) datasets. For the former group of datasets, we use one-hot encoded degrees as initial node features, while for the latter group, we use the provided node labels as initial node features. We then pass the features, alongside the adjacency matrices, to three popular graph neural network (GNN) backbones, namely Graph Convolutional Network (GCN) [17], Graph Attention Network (GAT) [18], and Graph Isomorphism Network (GIN) [19]. Upon deriving the final node embeddings of a given graph from a GNN backbone, we treat them as elements of a set and apply PSWE and the baseline pooling methods to derive a fixed-size graph-level representation that is fed to a linear classifier.

Table 2 shows the resulting 10-fold cross-validation accuracies on different datasets, and using different backbone/pooling pairings, following the evaluation methodology used in the literature [19, 50, 51]. As the table demonstrates, PSWE is able to perform similarly to or better than other pooling methods on all datasets. Furthermore, the results show that the commonly used mean-pooling for GNNs might not be the best choice, and more complex backbones might be needed to enhance the classification performance. It is important to note that in some scenarios, especially with smaller datasets (e.g., IMDB-B / ENZYMES) and more complex backbones (e.g., GIN), the performance of PSWE does not monotonically improve with the number of slices, $L$, which is due to the model becoming more complex and overfitting the training data.

| Dataset | Backbone | MAX | GAP | FSPool | PSWE |
|---------|----------|-----|-----|--------|------|
| **NWPU-RESISC45** | $16 \times 16$ Patches + MLP | $71.0 \pm 0.3$ | $65.0 \pm 0.7$ | $77.2 \pm 0.6$ | $75.0 \pm 0.9$ |
| | ResNet18 | $89.3 \pm 0.5$ | $91.4 \pm 0.4$ | $90.7 \pm 0.3$ | $90.6 \pm 0.5$ |
| **Places-Extra69** | $16 \times 16$ Patches + MLP | $18.5 \pm 2.2$ | $29.1 \pm 0.1$ | $35.4 \pm 1.8$ | $35.2 \pm 0.2$ |
| | ResNet18 | $57.3 \pm 0.1$ | $58.6 \pm 0.0$ | $57.3 \pm 1.8$ | $58.3 \pm 0.0$ |

Table 3: Image classification results (% test accuracy) on the NWPU-RESISC45 and Places-Extra69 datasets using two backbone types coupled with MAX, GAP, FSPool, and PSWE pooling methods.

### 5.3 Image Recognition

Finally, we evaluate PSWE in the context of image recognition on two large-scale image datasets: NWPU-RESISC45 [46], which is a remote sensing image scene classification dataset comprising a total of 31,500 images belonging to 45 different aerial scene classes, and Places-Extra69 [47], which contains 98,721 training and 6,600 test images, belonging to 69 different scene categories. For processing the images, we consider two different backbone types:

- **$16 \times 16$ Patches + MLP:** Inspired by the architecture used in the Vision Transformer (ViT) framework [52], we break the image into 256 patches, each flattened into a $16 \times 16 \times 3 = 768$-dimensional vector, pass each patch through a shared multi-layer perceptron (MLP), add positional encoding to the MLP outputs, and treat the outputs as a set of 256 elements, each with 256 features.

- **ResNet18 [53]**: As an alternative, we pass the image through ResNet18, which is a convolutional neural network backbone, mapping the input image into a $7 \times 7 \times 512$-dimensional tensor. We treat this tensor as a set of 49 elements, each containing 512 features.

Table 3 shows the test classification accuracy of PSWE, as compared to GAP, MAX, and FSPool on the two datasets using the two aforementioned backbones. For PSWE, we set the number of slices to $L = 1024$ for the $16 \times 16$ Patches + MLP backbone, and $L = 1000$ for the ResNet18 backbone. We did not include PMA results here as it performed significantly worse than other pooling types. As the table shows, PSWE performs on par with FSPool using both backbones, and significantly better than GAP with the simpler MLP-based backbone. This is consistent with our observation that more sophisticated pooling mechanisms can compensate the performance drop caused by simpler backbone architectures.

## 6 Conclusion

We introduced a novel method for permutation-invariant feature aggregation from set-structured data, called pooling by sliced-Wasserstein embedding (PSWE). Our method treats the elements of each input set as samples from a distribution, and derives a constant-size representation for the entire set based on the (generalized) sliced-Wasserstein distance between the set elements and a reference set, whose elements are learned in an end-to-end fashion, alongside with the slicer parameters. We showed that our method derives an exact Euclidean embedding which is geometrically-interpretable for set-structured data. Moreover, we demonstrated, through experimental results, that our set embedding approach outperforms baseline pooling mechanisms on a variety of supervised classification tasks on point cloud, graph, and image datasets. While our focus in this work was on deriving global representations for input samples (such as point clouds, graphs, and images), our method is not necessarily limited to global pooling. Indeed, our approach is a generic mechanism for embedding an input set into a fixed-dimensional representation and, therefore, it may also be used it for local pooling, which is an interesting direction for future work.

## Acknowledgments and Disclosure of Funding

This material is based upon work supported by the United States Air Force under Contract No. FA8750-19-C-0098. Any opinions, findings, and conclusions or recommendations expressed in this material are those of the authors and do not necessarily reflect the views of the United States Air Force and DARPA.

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
