# Pooling by Sliced-Wasserstein Embedding: Supplementary Material

**Navid Naderializadeh**[*]
Department of Electrical and Systems Engineering
University of Pennsylvania
Philadelphia, PA 19104
nnaderi@seas.upenn.edu

**Joseph F. Comer, Reed W. Andrews, Heiko Hoffmann**
HRL Laboratories, LLC.
Malibu, CA 90265
{jfcomer, rwandrews, hhoffmann}@hrl.com

**Soheil Kolouri**[*]
Computer Science Department
Vanderbilt University
Nashville, TN 37235
soheil.kolouri@vanderbilt.edu

## 1 Proof of C1 and C2 in Section 4.1

First we show that the reference is mapped to the origin, i.e., $\nu_0^\theta = \mathbf{0}$.

$$\nu_0^\theta = F_{\mu_0^\theta}^{-1} \circ F_{\mu_0^\theta} - id = id - id = \mathbf{0},$$

where we used Monge map formulation in (7) and the CDT definition in (8). Now, we have

$$\begin{aligned}
\|\nu_i^\theta - \nu_j^\theta\|_{\mu_0^\theta, 2} &= \|f_i^\theta - f_j^\theta\|_{\mu_0^\theta, 2} \\
&= \left( \int_{\mathbb{R}} \|f_i^\theta(t) - f_j^\theta(t)\|^2 d\mu_0^\theta(t) \right)^{\frac{1}{2}} \\
&= \left( \int_{\mathbb{R}} \|F_{\mu_i^\theta}^{-1}(F_{\mu_0^\theta}(t)) - F_{\mu_j^\theta}^{-1}(F_{\mu_0^\theta}(t))\|^2 d\mu_0^\theta(t) \right)^{\frac{1}{2}} \\
&= \left( \int_0^1 \|F_{\mu_i^\theta}^{-1}(u) - F_{\mu_j^\theta}^{-1}(u)\|^2 du \right)^{\frac{1}{2}} \\
&= \mathcal{W}_2(\nu_i^\theta, \nu_j^\theta),
\end{aligned}$$

which completes the proof of C2. Finally, given C2 and the fact that $\nu_0^\theta = \mathbf{0}$, we can write

$$\|\nu_i^\theta\|_{\mu_0^\theta, 2} = \|\nu_i^\theta - \nu_0^\theta\|_{\mu_0^\theta, 2} = \mathcal{W}_2(\nu_i^\theta, \nu_0^\theta),$$

which completes the proof of C1.

---

[*]Work done while at HRL Laboratories, LLC.

35th Conference on Neural Information Processing Systems (NeurIPS 2021).

## 2 Implementation Details

### 2.1 License

The CAD models in ModelNet40 [1] were downloaded from the Internet and labeled using Amazon Mechanical Turk (MTurk). The original authors hold the copyright of the CAD models. The TUD graph datasets [2] can be used under a Creative Commons Attribution-ShareAlike License. The NWPU-RESISC45 dataset [3] is available for use in accordance to the fair use exception to copyright infringement. The Places-Extra69 dataset [4] can be used under the Creative Common License, and the copyright of all the images belongs to the corresponding image owners. None of the datasets that we use for evaluation contains personally identifiable information or offensive content

### 2.2 Pooling Methods

We use a single-layer linear mapping as the slicer for all PSWE experiments (as opposed to a generalized non-linear slicer mechanism). For a fair comparison, we set the number of attention heads of Pooling by Multi-head Attention (PMA) [5] to 1. For experiments where the cardinalities of all sets are equal (i.e., point cloud and image experiments), we use that as the size of the reference set in PSWE and the number of seeds in PMA. On the other hand, for graph experiments, we set the size of the reference set in PSWE and the number of seeds in PMA to the average number of graph nodes in the training dataset.

### 2.3 Point Cloud Processing

For the training objects, we first sample $2048$ points uniformly at random from each object, and then randomly take $1024$ of those samples in each training epoch, while we fix the set of $1024$ samples for each of the test objects. We use random scaling (chosen uniformly at random from $[\frac{2}{3}, \frac{3}{2}]$), random translation (chosen uniformly at random from $[-0.2, 0.2]$), and random rotation in the $x$-$y$ plane (chosen uniformly at random from $[-30°, 30°]$) as augmentations for point clouds in the training dataset.

We use two types of backbones:

- **MLP:** We consider a multi-layer perceptron (MLP) with two 256-dim hidden layers, and a 256-dim output layer, which independently maps the initial three features of each element in the point cloud to a 256-dim embedding. We use rectified linear unit (ReLU) non-linearity after each of the two hidden layers.

- **ISAB:** We consider the induced set attention block (ISAB) backbone [5], consisting of two 256-dim ISAB layers, each with 4 attention heads and 16 inducing points. Through this backbone, the elements within each point cloud set perform attention-based message passing to map their initial three features to 256-dim embeddings.

We set the batch size to 32, and train each configuration, i.e., (backbone, pooling) pair, using Adam optimizer for 200 epochs. We set the initial learning rate to $10^{-3}$ and decay it by $0.5$ every 50 epochs. We run all experiments with 10 different random seeds and report the mean and standard deviation of the test accuracies across those seeds in Table 1.

### 2.4 Graph Classification

For the bio-informatics datasets (i.e., ENZYMES, PROTEINS), we use the provided node attributes as the initial node features. Moreover, for the social network datasets (i.e., IMDB-B, IMDB-M, REDDIT-B), we respectively use 300-dim, 300-dim, and 500-dim one-hot encoded node degrees as the initial node features, and clip the node degrees if they are beyond the aforementioned dimensions.

As for the backbones, we consider three different GNN backbones, namely Graph Convolutional Network (GCN) [6], Graph Attention Network (GAT) [7], and Graph Isomorphism Network (GIN) [8], where each backbone has a distinct type of message passing mechanism among the graph nodes over the graph edges. We consider three 256-dim GNN layers (two hidden and one output) for each GNN type, where a ReLU non-linearity is used after each GNN layer except for the output layer. For GAT, we consider a single attention head, and for GIN, the neural network at each layer is set to an MLP

| Phase | Backbone | GAP | PMA | FSPool | PSWE | | | | | |
|---|---|---|---|---|---|---|---|---|---|---|
| | | | | | $L = 1$ | $L = 4$ | $L = 16$ | $L = 64$ | $L = 256$ | $L = 1024$ |
| Training | MLP | $21.35 \pm 0.09$ | $26.58 \pm 0.12$ | $22.82 \pm 0.06$ | $21.89 \pm 0.02$ | $21.91 \pm 0.02$ | $21.95 \pm 0.02$ | $22.33 \pm 0.02$ | $23.79 \pm 0.09$ | $30.17 \pm 0.24$ |
| | ISAB | $26.27 \pm 0.03$ | $30.50 \pm 0.11$ | $27.31 \pm 0.03$ | $26.71 \pm 0.02$ | $26.74 \pm 0.02$ | $26.76 \pm 0.02$ | $27.03 \pm 0.03$ | $28.06 \pm 0.04$ | $33.99 \pm 0.05$ |
| Inference | MLP | $0.16 \pm 0.00$ | $0.60 \pm 0.00$ | $0.40 \pm 0.00$ | $0.17 \pm 0.00$ | $0.17 \pm 0.00$ | $0.19 \pm 0.00$ | $0.27 \pm 0.00$ | $0.54 \pm 0.00$ | $1.68 \pm 0.00$ |
| | ISAB | $0.51 \pm 0.00$ | $0.97 \pm 0.00$ | $0.76 \pm 0.00$ | $0.52 \pm 0.00$ | $0.53 \pm 0.00$ | $0.54 \pm 0.00$ | $0.62 \pm 0.00$ | $0.89 \pm 0.00$ | $2.04 \pm 0.01$ |

Table 1: Comparison of per-epoch training and inference wall-clock times (in seconds) between PSWE and the baseline pooling methods on the ModelNet40 dataset.

with a single hidden layer and ReLU non-linearity, where the size of the hidden layer equals the sum of the input and output dimensions.

We set the batch size to 32, and train each configuration, i.e., (backbone, pooling, dataset) tuple, using Adam optimizer for 20 epochs. We fix the learning rate at $10^{-3}$. Following [8, 9, 10], we perform 10-fold cross-validation and report the mean and standard deviation of the validation accuracies across the 10 folds in Table 2.

## 2.5  Image Recognition

We consider two backbone types:

- **16 × 16 Patches + MLP:** We use random resized crop to map each input image to $256 \times 256$ pixels, and then partition the image to $16 \times 16$ patches, each of size $16 \times 16$ pixels. We then flatten the patches and pass each of them through a shared MLP. The MLP has two 256-dim hidden layers, each with batch-norm and Leaky-ReLU non-linearity, and a linear 256-dim output layer. At the MLP output, we add Fourier-based positional encoding [11] to derive the final 256-dim patch embeddings.
- **ResNet18**: We use random resized crop to map each input image to $224 \times 224$ pixels, and then pass the image through a ResNet18 architecture [12] to derive a $7 \times 7 \times 512$-dimensional tensor at its output, which we treat as a set of 49 elements, each containing 512 features. This implies that the average pooling module and the fully-connected layer at the end of the ResNet18 architecture are entirely removed.

Regardless of the backbone type, for the NWPU-RESISC45 dataset, we consider a sequence of random horizontal flip, random vertical flip, and random cutout as training augmentations, while for the Places-Extra69 dataset, we augment training images via a sequence of random horizontal flip and random cutout. We set the batch size to 32, and train each configuration, i.e., (backbone, pooling, dataset) tuple, using Adam optimizer for 100 epochs. We set the learning rate to 0.025, run all experiments with 3 different random seeds and report the mean and standard deviation of the test accuracies across those seeds in Table 3.

## 2.6  Hardware

We run our experiments on an internal cluster containing 18 nodes, each equipped with one of the following three CPU/GPU configurations:

(i)  2.40 GHz Intel® Xeon® E5-2680 v4 CPU and two 16 GB NVIDIA® Tesla® P100 GPUs.

(ii)  2.40 GHz Intel® Xeon® E5-2680 v4 CPU and four 16 GB NVIDIA® Tesla® P100 GPUs.

(iii)  2.90 GHz Intel® Xeon® Gold 6226R CPU and two 32 GB NVIDIA® Tesla® V100S GPUs.

## 3  Wall-Clock Time Comparison

Table 1 shows the per-epoch training and inference times of different pooling approaches on the ModelNet40 dataset. All experiments were run in isolation on a machine with the hardware configuration (iii) mentioned in Section 2.6.

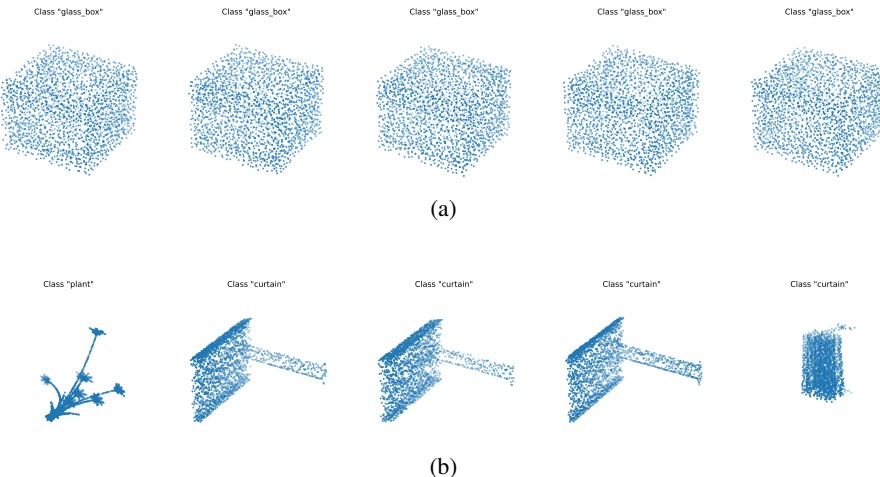

(a)

(b)

Figure 1: Training samples that are (a) closest to, and (b) farthest from the trained reference, after an ISAB backbone and a PSWE pooling module with $L = 1024$ are trained on the ModelNet40 dataset.

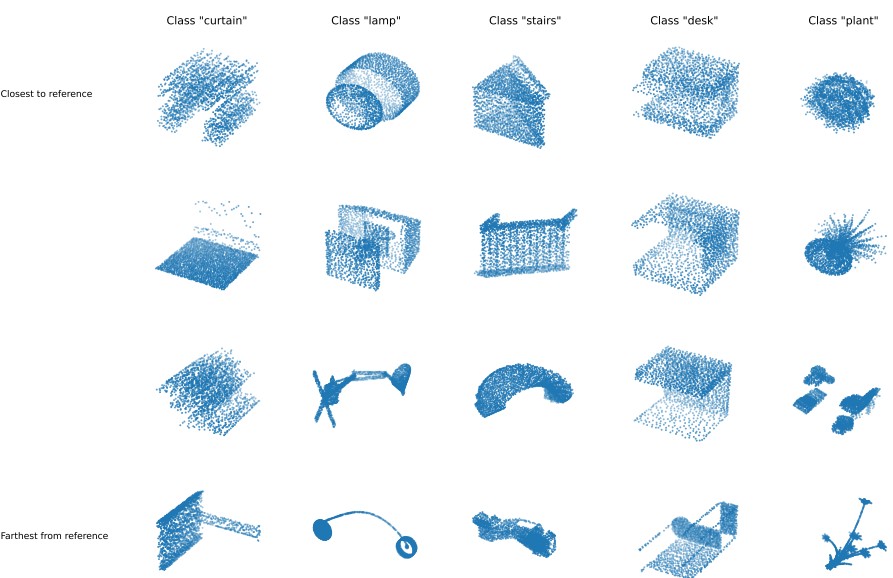

Figure 2: Visualization of samples from five classes in the ModelNet40 training dataset with the largest variance in the distance to the reference, after an ISAB backbone and a PSWE pooling module with $L = 1024$ are trained on the ModelNet40 dataset. Each column shows a specific class. The samples on the top row are the ones closest to the reference within the corresponding classes (with the most typical shapes), as opposed to the samples on the bottom row, which are the farthest samples from the reference in the corresponding classes (with the most atypical shapes).

## 4   Visualizing Closest and Farthest Samples to the Trained Reference

For the ModelNet40 dataset with an ISAB backbone and PSWE ($L = 1024$) pooling, we seek to visualize the closest and farthest training samples to/from the reference (after slicing) once training is complete. As shown in Figure 1, the closest samples to the reference all look like regular cubes and belong to the class "glass_box," while the farthest samples from the reference have very atypical shapes (e.g., "plants" with long branches). Moreover, Figure 2 illustrates a range of samples belonging to five classes which exhibited the largest variance in distances of their samples to the reference. As the figure shows, for a given class, as we get farther from the reference, the samples go from typical, prototype-like samples to the most atypical ones.