# OpenReview forum: "Pooling by Sliced-Wasserstein Embedding"
_NeurIPS.cc/2021/Conference — NeurIPS 2021 Poster_

### Official Review · Reviewer_iGeu · 2021-07-05

**Rating:** 7
**Confidence:** 3

**Summary:**

The paper proposes a method for learning how to aggregate a set of vector representations into a single representation for the whole set. This has applications in many tasks such as the common case of image recognition or representation where a 3D feature maps is pooled across the spatial dimensions, while other applications include point clouds and graph learning. The proposed approach is theoretically grounded and consists of an approximation of the 2-Wasserstein Distance. Experimental results contain experiments in 3 tasks: graph classification, point cloud classification, and image recognition.

**Limitations And Societal Impact:**

not discussed

**Main Review:**

- the proposed approaches has many applications - beyond the ones included in the experimental validation
- this is a novel approach that is theoretically grounded
- the fact that the final embeddings are not in the form of a matrix NxL but of a vector 1xL is mentioned only in the supplementary material. The matrix is multiplied with a learnable matrix, then, rows are summed and the embeddings are compared with Euclidean distance. I believe this information need to appear in the main paper. This should introduce performance loss (compared to Frobenious norm on the matrix embeddings). The performance of this is not mentioned in the paper. Despite the fact that this would be a more costly approach, it is useful to see what are the losses introduced by this last part of summing along rows.
- In table 2, there is large performance drop in some of the cases for L=1024. This is not discussed in the paper. In my understanding, the larger the L the better the approximation. Is there some explanation for this?
- is the trainable reference set explainable in any sense? This set of vectors lives in the same space as the input vectors in the considered sets. For instance, in the case of image recognition with 16x16 patches, similarity of each d dimensional vector in the reference with  patches from particular classes of the training set can be used to see association of these vectors with different classes. This as an illustrative example, but probably many more.

-----
post rebuttal:
I am still positive about the paper. The authors have provided detailed explanations which are very satisfactory. It is true that in some of the cases the performance is very close to that of GAP which is a simple as it gets. In some of the experiments the improvement over GAP is very large, while in others non-existing - some elaborated explanation of this discrepancy is missing. This is an aspect to improve in the camera ready.

**Time Spent Reviewing:**

5

---

> ### Author Response · Authors · 2021-08-10
> **Response to Reviewer iGeu**
>
> First, we would like to genuinely thank you for your service to the community as a reviewer amidst these troubling times, your constructive feedback, and your positive evaluation of our paper. Below we provide specific answers to your thoughtful comments.
>
> * **$N \times L$ versus $1 \times L$ embedding.** We agree with the reviewer that this is an important implementation detail, and we will move this information into the main body of the manuscript in the camera-ready version of the paper. We had carried out both implementations ($N \times L$ and $1 \times L$). The former one can fit the data better and incur a lower training loss, but one must be careful with such an implementation, as it might overfit the training data (which we actually observed in many of our experiments), and that is why we chose to go with the weighted summation over the $N$ elements of the reference set to achieve better generalizability.
>
> * **Explanation for performance drop for higher $L$s.** The culprit for the performance drop for higher $L$ is overfitting. This phenomenon can be especially observed in graph datasets with fewer training samples (e.g., IMDB-B / ENZYMES) and more complex backbones (e.g., GIN). We will explain this in the camera-ready version of the manuscript.
>
> * **Explainability of the optimized reference.** First, as pointed out in line 220, we optimize the reference in the embedding space as we found this to lead to a more well-behaved optimization landscape and ultimately better classification performance. This choice occludes explainability, but we can visualize and explain the reference directly if the optimization is performed in the input space. That said, following the reviewer's question, we sought to visualize the closest and farthest samples from the reference (after slicing). It turns out that, for the ModelNet40 dataset, the closest samples to our reference all looked like regular cubes and belonged to the class "glass_box," while the farthest samples from the reference had very atypical shapes (e.g., 'plants' with long branches). In general, for a given class, we observe that as we get farther from the reference, the samples go from typical, prototype-like samples to the most atypical ones. We will include visual results in the Supplementary Material of the camera-ready version of the paper. Another interesting avenue with respect to explainability is regarding the number of reference sets. Here, we only use and optimize a single reference set, but if explainability is the core focus, we can use multiple reference sets (analogous to multi-head attention in the PMA module of the Set Transformer architecture [14]) and visualize the (closest samples to each of the) optimized references.

---

> > ### Comment · Reviewer_iGeu · 2021-08-26
> > **authors response is appreciated**
> >
> > Thank you for the detailed explanations which are very satisfactory. I am still positive about this work. It is true, though, that in some of the cases the performance is very close to that of GAP which is a simple as it gets. In some of the experiments the improvement over GAP is very large, while in others non-existing - some elaborated explanation of this discrepancy is missing.

---

> > > ### Author Response · Authors · 2021-08-26
> > > **Thank you!**
> > >
> > > We appreciate the reviewer for their positive evaluation of our work and our response to their comments. As you alluded to, our experiments consistently show that the performance difference between PSWE and other methods is largest for shallower backbones and smallest for deeper backbones. This can be associated with the fact that the set distribution for shallower backbones cannot be simply described with its first moment and requires capturing higher moments, hence PSWE shines. On the other hand, deeper backbones (such as ResNet18) are capable of warping the space enough to simplify the set distributions such that they can be described via their first moment (e.g., their mean), and there will not be a need for analyzing higher moments.
> > >
> > > We will add an experiment to the camera-ready version of the paper, in which we visualize the output trained set distributions (before pooling) with different backbone types to confirm our hypothesis regarding the difference in the complexity of the set distributions with shallow vs. deep backbone architectures.

---

### Official Review · Reviewer_xtA8 · 2021-07-17

**Rating:** 5
**Confidence:** 3

**Summary:**

The paper proposes a pooling mechanism for embedding sets of various sizes into a fixed-size representation with a novel and theoretically-grounded pooling approach. The proposed Pooling by Sliced-Wasserstein Embedding (PSWE) is simple to compute and permutation-invariant. This pooling provides an exact Euclidean embedding for the generalized sliced-Wasserstein distance. Experimental results show that this pooling can be used in different kinds of data to boost performance, when comparing with basic pooling mechanism on plain backbones.

**Ethical Concerns:**

The proposed method is a general pooling technique, so I think there is no ethical concern involved.

**Limitations And Societal Impact:**

Yes. See Checklist 1.(c).

**Main Review:**

Originality:
	The proposed Pooling by Sliced-Wasserstein Embedding is novel for permutation-invariant feature aggregation from set-structured data. The proposed method seems based on this following paper

Generalized Sliced Wasserstein Distances, arXiv 2019.

And this work is an extension of the implementation/realization of a pooling module.

Quality:
Strengths:
 -  The theory is clearly defined and proven. Section 3 introduces the previous work on theorical background. Section 4 described the proposed pooling method based on Section 3 in dealing with the two considered scenarios and overall algorithm.

Weaknesses:
 - The result of the Image Recognition experiment and the result of the PSWE pooling only improve the performance by a little when compared to Global Average Pooling (GAP).
 - For the comparison of basic pooling methods, I suggest the authors to add more methods such as maximum pooling that are frequently used for comparison (in Table 3).

Clarity:
Strengths:
 - The paper is well-organized in introducing the basic theory and background to the description of the proposed method.
Weaknesses
 - Some of the symbols in the paper is confusing and do not clearly explain those introduced in the previous papers.
 - It is hard to re-implement without more detail about the modification of the model in the Image Recognition, e.g. which layer is modified.
 - In Line 103, the symbol d, i were not explained which I guess is the channel size of the feature map and hidden layer number i.
 - In Line 159, the symbol f_i^θ was not explained.

Significance:
The result shows the proposed PSWE pooling obtains better performance in the Point Cloud Classification and Graph Classification task. However there is not much distinguishable improvement in the Image Classification task.


**Time Spent Reviewing:**

3

---

> ### Author Response · Authors · 2021-08-10
> **Response to Reviewer xtA8**
>
> We would like to genuinely thank you for your service to the community as a reviewer amidst these troubling times and your constructive feedback. Below we provide specific answers to your thoughtful comments, which we hope will enable you to increase your rating.
>
> First, we point out that our paper is not really an extension of the GSWD paper [39]. Here, our focus is on providing a fixed-size Euclidean embedding for GSWD, similar to the general idea of linear Wasserstein embedding as used in the "Wasserstein Embedding for Graph Learning" paper [16]. The focus of our work, as correctly pointed out by the reviewer, is on learning from sets with potentially different cardinalities. We use a neural network backbone to map an input sample (e.g., image, graph, or point-cloud) into an empirical distribution (e.g., ResNet18 maps an input image of size $256\times 256$ into $7\times7$ samples in a $512$-dimensional space, which we treat as an empirical measure with $49$ samples in $\mathbb{R}^{512}$). We then use our pooling to embed this empirical distribution into a fixed-dimensional Euclidean embedding in which the Euclidean distance between two vectors corresponds to the GSW metric between the corresponding empirical distributions.
>
> * **Image Recognition results and comparison with maximum pooling.** Following your constructive feedback, we have performed a new set of image recognition experiments on the Places-Extra69 dataset with the "$16 \times 16$ Patches + MLP" backbone. In particular, we resolved a few minor issues with the image augmentations used during training and testing, and we added the max-pooling results as you suggested. Below please find the updated results of our experiment. As the table demonstrates, our proposed PSWE pooling method performs similarly to the FSPool method, while significantly outperforming global average-pooling (GAP) and global max-pooling (MAX). We will re-run the entire set of image recognition experiments in Table 3 of the paper (with max-pooling added), and we will add the updated results to the camera-ready version of the manuscript.
>
>     |        GAP       |    MAX    |   FSPool   |    PSWE|
>     |------------------|-----------------|-----------------|-----------------|
>     | 29.1 $\pm$ 0.1    | 18.5 $\pm$ 2.2    | 35.4 $\pm$ 1.8 | 35.2 $\pm$ 0.2 |
>
>
> * **Clarity and symbols.** We regret to see that we have missed some definitions of our symbols, and we thank you for pointing this out. We clarify that:
>     * $d$ denotes the dimensionality of each element of the input set
>     * $i$ denotes the index of an input set, i.e., $X_i$ is the $i^{th}$ input set, which will be treated as an empirical measure $\mu_i$.
>     * $f_i^\theta$ denotes the Monge map calculated between the slice of the $i^{th}$ measure $\mu_i$, where the slice is parameterized by $\theta$, and slice of the reference measure $\mu_0$.
>
>     We will clarify and describe all symbols in our updated manuscript.
> * **Implementation details.** Our framework could replace any global pooling mechanism, e.g., the global average pooling in ResNet. We have provided our code (for the point cloud processing experiments) in the supplementary material for your consideration. Also, we will release our code shortly after the decision period. Furthermore, following your suggestion, we will revise our Supplementary Material to better explain the implementation details.
>
> Again we hope that we have addressed your main concerns, and we would be happy to discuss any further questions you may have.

---

> > ### Author Response · Authors · 2021-08-11
> > **Updated Places-Extra69 Image Recognition Results with the ResNet18 Backbone**
> >
> > We have also re-run the image recognition experiments on the Places-Extra69 dataset with the ResNet18 backbone, taking into account the changes we mentioned in our previous comment (especially by modifying the augmentation process), and the results are shown in the table below.
> >
> > |        GAP       |    MAX    |   FSPool   |    PSWE|
> > |------------------|-----------------|-----------------|-----------------|
> > | 58.63 $\pm$ 0.03    | 57.31 $\pm$ 0.06    | 57.30 $\pm$ 1.80  | 58.30 $\pm$ 0.02   |
> >
> > As the table demonstrates, with this backbone, both PSWE and GAP are able to outperform FSPool and max-pooling, which shows the effectiveness of the proposed PSWE pooling method when used in conjunction with different backbone types and complexities.

---

> ### Comment · Reviewer_xtA8 · 2021-08-26
> **Minor review comments**
>
> After reading all other reviews and author feedback, I would like to increase my rating to 6: Marginally above the acceptance threshold, as the proposed method is really novel. However just one concern about the results:  results on 16x16 patches + MLP has improved greatly, however on resnet18 the improvement is not that much; so it seems a bit schematic.

---

> > ### Author Response · Authors · 2021-08-26
> > **On the difference in performance gains with different backbones**
> >
> > We appreciate the reviewer for considering increasing their evaluation score toward acceptance. Our experiments consistently show that the performance difference between PSWE and other methods is largest for shallower backbones and smallest for deeper backbones. This can be associated with the fact that the set distribution for shallower backbones cannot be simply described with its first moment and requires capturing higher moments, hence PSWE shines. On the other hand, deeper backbones (such as ResNet18) are capable of warping the space enough to simplify the set distributions such that they can be described via their first moment (e.g., their mean), and there will not be a need for analyzing higher moments.
> >
> > We will add an experiment to the camera-ready version of the paper, in which we visualize the output trained set distributions (before pooling) with different backbone types to confirm our hypothesis regarding the difference in the complexity of the set distributions with shallow vs. deep backbone architectures.
> >
> > We would be happy to continue the discussion to address any remaining concerns you might have.

---

> > ### Author Response · Authors · 2021-09-09
> > **Follow-up**
> >
> > We would like to follow up on our discussion to resolve any remaining concerns you might have about our work, and we would appreciate it if you could please increase your rating as you indicated.

---

### Official Review · Reviewer_vGLg · 2021-07-23

**Rating:** 7
**Confidence:** 3

**Summary:**

In this paper, the authors propose a method to compare sets by using a probability measure between two embedded sets. This is achieved through the sliced Wasserstein distance (SWD) between their empirical distributions. The method provides theoretical guarantees to prove the validity of their proposed measures. The use of SWD for sets is demonstrated for point cloud classification, graph classification and basic image classification.

**Ethical Concerns:**

The work uses publicly available datasets for set based evaluation and comparisons. Moreover, the contribution is mainly mathematical and is therefore not expected to have significant ethical concerns.

**Limitations And Societal Impact:**

The work is more mathematical in nature and is not expected to have a significant societal impact.

**Main Review:**

The use of SWD for learning architectures on sets is demonstrated using the idea of obtaining Euclidean embedding for the probability measures. The main ideas used are relying on cumulative distribution transform to obtain the distance measure. This then allows definition of the generalized sliced Wasserstein (GSW) to be used to measure the distance between two slices. The challenge then is to obtain slicer functions. These are obtained for the whole dataset using a reference set X_0. The procedure then is that given a dataset and a reference set, a number of slicers that are permutation equivariant are learned. These are used to obtain optimal transport maps between the slices. These transport maps are concatenated to obtain the final permutation invariant embeddings.

Pros:
1) The use of probability measures to obtain pooled SW distance functions is novel and this is related to other pooling functions.
2) The method is theoretically grounded using the CDT
3) The method is demonstrated on a number of different settings ranging from point-cloud classification to image recognition

Cons:
1) The paper would be improved by obtaining a more detailed ablation analysis of their method
2) The method is focused on comparing different set based datasets for the various architectures such GAT, GCN using the proposed pooling methods and comparing against the other pooling functions. However, the method can also be seen as proposing a novel GSW distance metric. Thus comparison against other such methods such as the max-sliced Wasserstein distance and other GSW distances in their settings could also be useful.

In conclusion, I believe the work provides a novel enough contribution that is interesting and useful. I therefore have a positive view about the paper.

**Time Spent Reviewing:**

10

---

> ### Author Response · Authors · 2021-08-10
> **Response to Reviewer vGLg**
>
> First, we would like to genuinely thank you for your service to the community as a reviewer amidst these troubling times, constructive feedback, and positive evaluation of our work. Below please find our specific comments.
>
> 1. **Further ablation studies.** We have run extensive ablation studies on: 1) the impact of optimizing the reference set versus freezing an a priori selected reference, and 2) the impact of optimizing the slices versus using a frozen set of slices (e.g., identity as performed in FSPool [15]). We observe that the main gain from optimizing the references and slices happens when the backbone is shallow. Given a chance, we will add these ablation studies to the camera ready version of our work.
>
> 2. **The method can be seen as a novel GSW metric.** This is certainly an interesting view, which is worth further considerations. For clarity, we would like to point out that the role of the backbone in our framework is to map each sample in the input space to an empirical distribution in the embedding space. Therefore, unlike the original GSW paper, we are not interested in slicing the input distribution (i.e., the distribution from which our samples are drawn); but in our setting, each input sample is first transformed into an empirical distribution and then the (Generalized)Sliced-Wasserstein Embedding is calculated for this empirical distribution with respect to a reference measure. The concept of (Generalized)Sliced-Wasserstein Embedding (i.e., extension of CDT [42] to $d$-dimensions) is crucial in our setting to enable pooling from sets with potentially different cardinalities.

---

> > ### Comment · Reviewer_vGLg · 2021-09-02
> > **Response**
> >
> > The response from the authors was useful in clarifying the points. I retain a positive view of the paper and would like to retain my accept rating

---

> > > ### Author Response · Authors · 2021-09-09
> > > **Thank you!**
> > >
> > > Thank you very much for your positive evaluation of our work and our responses to your comments.

---

### Official Review · Reviewer_SDvh · 2021-07-27

**Rating:** 6
**Confidence:** 4

**Summary:**


Summary
=======

Context:
--------

This paper tackles the deep learning setting where samples do not belong to a fixed dimensional space but
each sample is a set of potentially different cardinality. Since samples are sets, constructing learning or embedding functions
in this setting must be invariant to the order of their elements. The focus in this paper is on deep learning, specifically pooling layers
for set samples.


Contribution
------------

The authors propose to model each set (data observation x_i with cardinality N_i) as samples (x_i) draw from some distribution \mu_i.
Then, the authors show how to construct an embedding (pooling) function f that takes a measure \mu_i as input such that the euclidean distance
in the embedding space approximates the Sliced Wasserstein distance between the measures (sets):

                              \| f(\mu_i) - f(\mu_j) \| ~ SlidedWasserstein(\mu_i, \mu_j)

The output of the function f(\mu_i) is a vector \nu_i \in R^[N, L] where L is the number of slices used and N is the cardinality of a reference
set that is learned. The directions of the L slices are also learned during training.

The authors illustrate the performance of their pooling layer on various experiments showing  near state of the art performance specially for large L.



**Main Review:**

review
======

General assessment and main concerns
-----------------------------------
This paper is very well written and illustrated. The idea of using the 1D Monge map as an embedding output to preserve the GSW distances
in the output layer is brilliant, even though the experimental results are not significantly outperforming the baselines.

My main concerns are:

A) Given that the experiments have all similar results, I was left unsatisfied with regards to how crucial the role played by OT is.
Performing controlled simulated experiments could have answered questions like this one (+ the comments below) by controlling for e.g the dimension
of the elements of the set and computing the exact OT vs different values of L vs an MMD based network.

B) The geometrical motivation behind the work (as mentioned in the abstract and introduction). I'm afraid I don't see how
"geometry" plays any role here. While it certainly could in certain applications, the geometrical argument was not exploited nor motivated enough in this
paper to be a convincing argument. It seems like the Wasserstein distance was taken merely for: (1) the ability to compare measures regardless of the number
of samples; (2) the use of 1D slices to use fast argsort operations and have 1D couplings that preserve the distance in the embedding space. Given that The geometry here
(the ground metric) was set to L2 regardless of the nature of the data. So i'm wondering whether using a better informed (or simply a different) cost could improve / degrade
 the obtained comparisons.


Specific comments
-----------------

1) The introduction should provide a formal statement of the context (each observation is a set with cardinality N_i ..); a permutation invariant function
should verify ... I found the first read of this paper quite difficult without going through the Deep Sets paper again.

2) If I understand properly, the reference measure could be fixed to any arbitrary measure. Does learning it improve the performance of the model
by a significant margin ?

3) Since it is learned here for optimal results, could it be interpreted or visualized in the context of images for instance ?

4) Performances being equal, how does the proposed method compare in terms of run time ?

minor:
5) The proposed method has an output in R^(N, L) which is likely to have more elements than the input set. Moreover, it learns
an embedding that preservers GSW distance across sets. This is more likely to correspond to an entire network than a "pooling"
operation.


**Time Spent Reviewing:**

3

---

> ### Author Response · Authors · 2021-08-10
> **Response to Reviewer SDvh**
>
> First we would like to genuinely thank you for your generous service to the community as a reviewer amidst these troubling times, and for your constructive feedback. Below please find our point-to-point responses to your comments.
>
> ### Main concerns
> **A. Performing controlled simulated experiments:** We agree with the reviewer that our paper has room for improvement concerning the addition of controlled experiments. Given a chance, we will add these experiments to the camera-ready version of the paper. Regarding direct comparison with OT, we tried the Optimal Transport Kernel Embedding (OTKE) framework proposed by Mialon et al. [12] in ICLR 2021; however, this approach is computationally expensive. Given the extensive number of experiments in our paper, we could not afford to run OTKE. We will provide the comparison on smaller-scale controlled experiments. Regarding MMD, it is unclear whether one can define a linearized MMD embedding with respect to a reference, and we will be happy to learn more about the possibilities here.
>
> **B. Geometrical motivation unclear/unexploited:** We completely agree with the reviewer that we could have done a better job describing the geometric picture of our proposed pooling. We will correct this in the camera-ready version of the paper. We would like to emphasize that the purpose of slicing in our work is not just to speed up the computation as is often the case for (generalized) sliced-Wasserstein distances, and but also due to the geometric motivations. First, while we are using the $\ell_2$ ground metric, it is worth noting that this distance is used on the slices (i.e., the marginal probability measures). In other words, when optimizing the set of slices with respect to an objective, we no longer use the $\ell_2$ ground metric in the input space but a subspace distance. In particular, for an optimized slice (i.e., max-sliced-Wasserstein), the ground metric in the input space becomes insensitive to variations along hyperplanes orthogonal to the projection vector (therefore, it is not $\ell_2$ anymore). We will add similar explanations to the paper to increase our work's clarity and further emphasize the geometric motivation behind our proposed pooling method.
>
> ### Specific comments
> 1. Thank you for the constructive comment. We will update our introduction to state our work's objective formally. Also, we will add more background on learning from sets to make our work standalone.
> 2. Yes, the reference and the slices could be frozen. In fact, FSPool [15] could be thought of as a special case of our PSWE, with a frozen reference set and a frozen set of slices (i.e., identity projections). We ran more extensive ablation studies and we will add the results to the Supplementary Material in the camera-ready version of the paper.
> 3. First, as pointed out in line 220, we optimize the reference in the embedding space as we found this to lead to a more well-behaved optimization landscape and ultimately better classification performance. This choice occludes explainability, but we can visualize and explain the reference directly if the optimization is performed in the input space. That said, following the reviewer's question, we sought to visualize the closest and farthest samples from the reference (after slicing). It turns out that, for the ModelNet40 dataset, the closest samples to our reference all looked like regular cubes and belonged to the class "glass_box," while the farthest samples from the reference had very atypical shapes (e.g., 'plants' with long branches). In general, for a given class, we observe that as we get farther from the reference, the samples go from typical, prototype-like samples to the most atypical ones. We will include visual results in the Supplementary Material of the camera-ready version of the paper. Another interesting avenue with respect to explainability is regarding the number of reference sets. Here, we only use and optimize a single reference set, but if explainability is the core focus, we can use multiple reference sets (analogous to multi-head attention in the PMA module of the Set Transformer architecture [14]) and visualize the (closest samples to each of the) optimized references.
>
> 4. Regarding wall-clock time comparison, we ran extensive experiments and below we include the per-epoch training and inference times of different pooling approaches on the ModelNet40 dataset. All experiments were run in isolation on a $2.90$ GHz Intel&reg; Xeon&reg; Gold 6226R CPU and two $32$ GB NVIDIA&reg; Tesla&reg; V100S GPUs.
>
>    **Average train time per epoch (seconds)**
>
>     | Backbone |        GAP       |    PMA    |   FSPool   |    PSWE (L=1)   |    PSWE (L=4)   |   PSWE (L=16)   |   PSWE (L=64)   |   PSWE (L=256)  |  PSWE (L=1024)  |
>     |----------|------------------|-----------------|-----------------|-----------------|-----------------|-----------------|-----------------|-----------------|-----------------|
>     |   MLP    | 21.35 $\pm$ 0.09 | 26.58 $\pm$ 0.12 | 22.82 $\pm$ 0.06 | 21.89 $\pm$ 0.02 | 21.91 $\pm$ 0.02 | 21.95 $\pm$ 0.02 | 22.33 $\pm$ 0.02 | 23.79 $\pm$ 0.09 | 30.17 $\pm$ 0.24 |
>     |   ISAB   | 26.27 $\pm$ 0.03 | 30.50 $\pm$ 0.11 | 27.31 $\pm$ 0.03 | 26.71 $\pm$ 0.02 | 26.74 $\pm$ 0.02 | 26.76 $\pm$ 0.02 | 27.03 $\pm$ 0.03 | 28.06 $\pm$ 0.04 | 33.99 $\pm$ 0.05 |
>
>
>     **Average test time per epoch (seconds)**
>
>     | Backbone |        GAP       |    PMA    |   FSPool   |    PSWE (L=1)   |    PSWE (L=4)   |   PSWE (L=16)   |   PSWE (L=64)   |   PSWE (L=256)  |  PSWE (L=1024)  |
>     |----------|------------------|-----------------|-----------------|-----------------|-----------------|-----------------|-----------------|-----------------|-----------------|
>     |   MLP    | 0.16 $\pm$ 0.00 | 0.60 $\pm$ 0.00 | 0.40 $\pm$ 0.00 | 0.17 $\pm$ 0.00 | 0.17 $\pm$ 0.00 | 0.19 $\pm$ 0.00 | 0.27 $\pm$ 0.00 | 0.54 $\pm$ 0.00 | 1.68 $\pm$ 0.00 |
>     |   ISAB   | 0.51 $\pm$ 0.00 | 0.97 $\pm$ 0.00 | 0.76 $\pm$ 0.00 | 0.52 $\pm$ 0.00 | 0.53 $\pm$ 0.00 | 0.54 $\pm$ 0.00 | 0.62 $\pm$ 0.00 | 0.89 $\pm$ 0.00 | 2.04 $\pm$ 0.01 |
>
>
> 5. While the proposed pooling method itself provides an embedding in  $\mathbb{R}^{N\times L}$, to avoid a very high-dimensional embedding, we take a weighted-sum of the emebddings over the first dimension (i.e., the $N$ elements of the reference set) so that the output is $L$-dimensional. This strategy is similar to the one used in FSPool. We will move this information from the Supplementary Material to the body of the paper for enhanced clarity.

---

> > ### Comment · Reviewer_SDvh · 2021-09-01
> > **response to authors**
> >
> > Thank you for the very detailed response. It cleared most of my concerns. I kept my score unchanged because I believe the experiments should meet higher standards. It is only equally / slightly better than the state of the art, conducting simulations could have shed some light on how sliced OT is actually useful here (the alternative being: any distance between measures could be equally good). Such experiments could have perhaps highlighted situations where the geometrical aspect of OT would be better suited than other pooling methods. Nonetheless, I still value this work very positively and encourage the authors to improve upon this submission.

---

> > > ### Author Response · Authors · 2021-09-09
> > > **Thank you!**
> > >
> > > Thank you very much for your positive evaluation of our work. We agree that the paper would benefit from more controlled experiments to show the contribution of sliced OT as compared to other distance metrics between measures (such as MMD), and if given a chance, we will add those experiments to the camera-ready version of the paper.

---

### Decision · Program_Chairs · 2021-09-28

**Decision:**

Accept (Poster)

**Comment:**

All the reviewers are positive about the work and also appreciated the rebuttal, where extra numerical experiments were promised, which I strongly encourage to add. Also an in-depth discussion of the variations in relative performances with respect to GAP seems important. As a small side note, the notion of SW was introduced in [38] and not [36, 37] as the end of Section 3.1 seems to imply (in particular [38] is not concerned with generalized SW).

**Consistency Experiment:**

NeurIPS has a long history of experimentation. In 2014, NeurIPS ran an experiment in which 10% of submissions were reviewed by two independent committees to quantify the randomness in the review process. This year, we repeated a variant of this experiment to see how the quality of the review process has changed over time.  This paper was part of the experiment and was therefore assigned to two committees (consisting of reviewers, an Area Chair, and a Senior Area Chair) that reached independent decisions.  If both committees made the same recommendation, this recommendation was followed. If a single committee recommended acceptance, the paper was accepted (with the exception of a few cases in which the other committee identified what we considered a fatal flaw, e.g., an error in a key result).

This copy’s committee reached the following decision: **Accept (Poster)**

The other committee assigned to the paper recommended **Reject**.  You can find the other set of reviews, along with any follow up discussion with the authors here:
https://openreview.net/forum?id=ienTaVMRtl